# Calcium/Calmodulin-Stimulated Protein Kinase II (CaMKII): Different Functional Outcomes from Activation, Depending on the Cellular Microenvironment

**DOI:** 10.3390/cells12030401

**Published:** 2023-01-23

**Authors:** John A. P. Rostas, Kathryn A. Skelding

**Affiliations:** 1School of Biomedical Sciences and Pharmacy, College of Health Medicine and Wellbeing, The University of Newcastle, Callaghan, NSW 2308, Australia; 2Hunter Medical Research Institute, New Lambton Heights, NSW 2305, Australia

**Keywords:** CaMKII, molecular targeting, binding protein, protein phosphorylation, calcium/calmodulin

## Abstract

Calcium/calmodulin-stimulated protein kinase II (CaMKII) is a family of broad substrate specificity serine (Ser)/threonine (Thr) protein kinases widely expressed in many tissues that is capable of mediating diverse functional responses depending on its cellular and molecular microenvironment. This review briefly summarises current knowledge on the structure and regulation of CaMKII and focuses on how the molecular environment, and interaction with binding partner proteins, can produce different populations of CaMKII in different cells, or in different subcellular locations within the same cell, and how these different populations of CaMKII can produce diverse functional responses to activation following an increase in intracellular calcium concentration. This review also explores the possibility that identifying and characterising the molecular interactions responsible for the molecular targeting of CaMKII in different cells in vivo, and identifying the sites on CaMKII and/or the binding proteins through which these interactions occur, could lead to the development of highly selective inhibitors of specific CaMKII-mediated functional responses in specific cells that would not affect CaMKII-mediated responses in other cells. This may result in the development of new pharmacological agents with therapeutic potential for many clinical conditions.

## 1. Introduction

Protein kinases regulate the function of proteins by phosphorylation at key locations where the negatively charged phosphate group induces conformational changes in the phosphorylated protein that alter its activity or ability to interact with other molecules. Some protein kinases have very narrow substrate specificity so they only regulate one, or very few, proteins and can only be activated in one way (e.g., binding of an ion such as Ca^2+^ or a small molecular weight molecule such as cAMP). Other protein kinases have a broad substrate specificity so they can regulate a large number of proteins and functional responses. Such kinases, often referred to as multifunctional kinases, usually have multiple mechanisms of activation and regulation.

Calcium/calmodulin-stimulated protein kinase II (CaMKII) is a family of multi-functional serine (Ser)/threonine (Thr) protein kinases (α, β, γ, and ∂), encoded by four genes (*CAMK2A*, *CAMK2B*, *CAMK2G*, and *CAMK2D*) [1]. One or more members of this family are expressed in almost every tissue, and control diverse functions.

CaMKII is expressed most highly in neurons and, consequently, its functional roles have been most extensively studied in neurons. The neuronal functions that CaMKII is critical in controlling include neurotransmitter synthesis and exocytosis, synaptic organisation, long-term plasticity, learning, memory consolidation and erasure following retrieval [2,3,4,5,6,7,8]. CaMKII also plays important roles in several neuronal pathologies, including impaired learning, transient focal or global ischaemia, excitotoxic neuronal cell death, Alzheimer’s disease, epileptogenesis, tissue-injury evoked persistent pain, and Parkinson’s disease [9,10,11,12,13,14,15,16,17,18,19,20]. Further, non-neuronal CaMKII has been implicated in regulating a variety of other cellular processes, including the cell cycle, fertilisation, cancer cell proliferation and metastasis, contraction-induced glucose uptake in skeletal muscle, insulin secretion, CD8 T cell function, intestinal motility, differentiation, cardiac function, and vascular tone [21,22,23,24,25,26,27,28,29,30,31,32].

Herein, we provide an overview of the structure and regulation of CaMKII, with a focus on how the molecular environment, particularly the interaction of CaMKII with binding proteins, can control the function of CaMKII.

## 2. CaMKII Structure

Each isoform of CaMKII exhibits the same basic domain structure: an N-terminal catalytic domain, a C-terminal association domain, and a regulatory domain in between (Figure 1A). The regulatory domain contains autoinhibitory and calcium/calmodulin binding regions, which partially overlap [33,34,35]. In the autoinhibited (inactive) state, the regulatory domain partially wraps around the catalytic domain (Figure 1B) and the autoinhibitory region acts as a pseudo-substrate, blocking the substrate binding site and inhibiting the kinase activity. The binding of calcium/calmodulin to the calcium/calmodulin binding region disrupts the interactions between the autoinhibitory region and the catalytic domain [34], causing the catalytic domain to rotate, opening the former compact structure (Figure 1C) and allowing the catalytic domain to phosphorylate substrates. Hence, the regulatory domain functions like a gate (with Thr286 acting like a ‘hinge’), so that it blocks the ATP and substrate binding sites when CaMKII is autoinhibited and ‘opens’ when CaMKII is activated by binding of calcium/calmodulin or by autophosphorylation at Thr286. Additionally, four main variable regions (V1–4) have been identified, which, through alternative splicing, can produce more than 30 isoforms, with the V1 region acting as the primary source for divergence among the four CaMKII genes [36].

The expression of CaMKII isoforms is highly variable across different tissues, and cells can contain more than one isoform. CaMKIIα and β are mainly expressed in nervous tissue [1], whereas γ and ∂ are expressed at low levels in virtually all cells [37]. Different isoforms can co-assemble into heteromeric holoenzymes [38], so both homo- and hetero-multimers of CaMKII presumably exist in vivo [39,40,41,42]. This can further fine tune CaMKII, as different CaMKII isoforms have different affinities for calmodulin [40], and also exhibit differing enzyme kinetics [43]. Taken together, this indicates that heteromeric CaMKII may exhibit differing functional properties depending on the isoform composition.

**Figure 1 cells-12-00401-f001:**
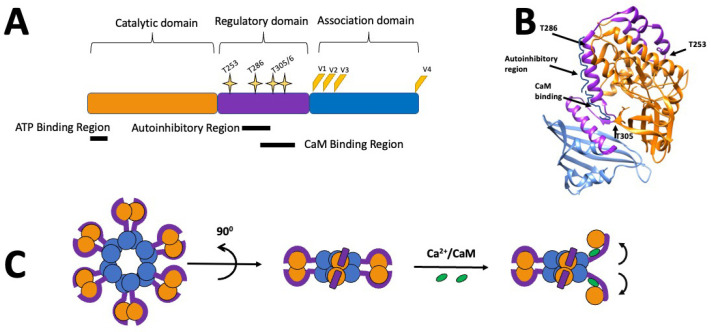
Schematic representations of the molecular structure of CaMKIIα: (**A**) CaMKII consists of a catalytic domain (orange), a regulatory domain (purple), which includes overlapping autoinhibitory and calmodulin (CaM) binding regions and multiple phosphorylation sites (yellow stars). The association domain (blue) is involved in the formation of CaMKII multimers. Splice sites (V1–4) are represented in orange/yellow. (**B**) Crystal structure (3SOa) of a single CaMKIIα subunit created using Chimera [44]. (**C**) Representation of the CaMKIIα holoenzyme structure, with the domains shown in the same colours as in (**A**), and showing the movement of the subunit and its catalytic domain following activation with calcium/calmodulin.

The majority of the CaMKII holoenzyme exists as a dodecamer consisting of two hexameric rings of subunits principally held together by the association domains which form an annulus with the association and catalytic domains forming two rings around the annular hub of association domains [45,46] (Figure 1C). The arrangement of the catalytic and regulatory domains around the hub of association domains appears to be in a dynamic equilibrium with the majority being in an extended state, in which the subunits are activatable by calcium/calmodulin, and the minority being in one of two compact states in which the subunits are not activatable [47]. A small proportion of holoenzymes also exist in a tetradecameric structure consisting of two heptameric rings of subunits [46]. When CaMKII is activated, the allosteric changes in the subunits weaken the interactions between the association domains in the annular hub and enable an interconversion of dodecameric and tetradecameric assemblies and an exchange of subunits between holoenzymes [48]. It has been suggested that such subunit exchange can spread activation in a population of CaMKII holoenzymes, some of which were not activated, and that this mechanism may be involved in information storage during neuronal signalling [49].

## 3. Regulation of CaMKII

CaMKII is regulated by multi-site phosphorylation and targeting to subcellular locations via interactions with specific binding proteins. Additionally, there is an interplay between these two regulatory mechanisms, as these interactions can be further modified by phosphorylation of both/either the kinase, and/or the binding protein. The interaction between the regulatory mechanisms of multi-site phosphorylation and molecular targeting through interaction with specific binding proteins, creates functionally different pools of CaMKII in different cells and different parts of the same cell. For this reason, the well-characterised behaviour of purified CaMKII in vitro often does not correctly predict the behaviour of CaMKII in vivo. It is also the existence of these functionally different pools of CaMKII that enables CaMKII to be a multi-tasking protein.

### 3.1. Autophosphorylation

All the CaMKII autophosphorylation sites are present in every CaMKII subunit and splice isoform and are conserved across species. We will review the functional effects of autophosphorylation at Thr286, Thr305/306 and Thr253, which are the three best-characterised sites.

#### 3.1.1. Thr286

Initial enzyme activity of purified CaMKII requires the binding of both calcium and calmodulin. Following this, autophosphorylation of Thr286 in CaMKIIα (Thr287 in CaMKIIβ, γ and ∂) occurs quickly and produces the following outcomes:A 1000-fold increase in the affinity for calcium/calmodulin is induced, thereby prolonging the calcium/calmodulin-stimulated activity of CaMKII due to calmodulin ‘trapping’, which results in calmodulin dissociating more slowly once free calcium returns to basal levels [50].Induction of autonomous activity, which allows the enzyme to remain active even following calmodulin dissociation. Calcium/calmodulin binding disrupts the interaction between the autoinhibitory domain and the ATP and substrate binding sites, thereby activating CaMKII (reviewed in [36,51,52]). Autophosphorylation of Thr286 requires calcium/calmodulin binding to adjacent subunits, which results in the trans-autophosphorylation of Thr286 [50]. Due to the combined effects of calmodulin ‘trapping’, autonomous activity and the requirement for activation of adjacent subunits, CaMKII can decode the duration, frequency and amplitude of calcium spikes and translate them into graded responses of kinase activity [53].Change the affinity of CaMKII binding to specific proteins that are located in specific subcellular sites (Table 1). This allows the translocation and targeting of CaMKII to specific cellular and molecular sites, which can vary between different cells depending on the CaMKII binding proteins they express, thereby selectively regulating downstream functions following CaMKII activation.

Under certain pathological conditions that result in periods of elevated reactive oxygen species, a pair of methionine residues in CaMKII (Met281/282), present in β, γ and δ, but not α isoforms, can become oxidised, resulting in an autonomous activation of CaMKII [54]. This persistent activation of CaMKII by oxidation of M281/282 has been shown to play an important role in cardiac myocyte cell death following ischaemia/reperfusion [54].

Recent biochemical evidence also suggests that autonomous activity triggers the colocalization of subunits between holoenzymes [48,49,55], and that CaMKII can transfer its activity to naïve CaMKII [56]. These findings would potentially account for how the brain can maintain memories that last longer than the proteins involved in their encoding.

#### 3.1.2. Thr305/306

Thr305/306 in CaMKIIα, and Thr306/307 in CaMKIIβ, γ and ∂, are a pair of secondary sites located within the calmodulin binding domain that can become phosphorylated once CaMKII is calcium-independent (autonomous) and can be phosphorylated under different conditions [57,58]. Phosphorylation at either or both of these sites prevents calmodulin binding (Table 1), thereby rendering CaMKII insensitive to changes in calcium/calmodulin [59]. Although Thr305/306 are often referred to as ‘inhibitory’ sites, this is a misnomer, as phosphorylation of these sites can only occur following Thr286 phosphorylation, and Thr305/306 phosphorylation does not alter autonomous activity. Surprisingly, Thr305/306 can also be phosphorylated in the absence of calcium/calmodulin at a slow rate in vitro [60]. However, whether CaMKII can become phosphorylated in the absence of calcium spikes in vivo is unknown.

#### 3.1.3. Thr253

Thr253 was the first characterised CaMKII autophosphorylation site that controls CaMKII activity purely by regulating molecular targeting and has no effect on the activity of purified CaMKII in vitro (Table 1). Initially, the phosphorylation of Thr253 was considered to be of questionable functional significance because, when measured in whole brain, the stoichiometry of Thr253 phosphorylation was low. However, Thr253 phosphorylation in vivo only occurs in small specific pools of CaMKII, such as the PSD where the phosphorylation stoichiometry is high [56], highlighting that the functions of Thr253 phosphorylation appear to be concentrated at specialised cellular locations, such as the PSD.

Other sites, such as Ser279 and Ser314, can be phosphorylated both in vitro [60,61,62] and in vivo [63,64,65], but like Thr253, the phosphorylation stoichiometry in whole brain is relatively low and phosphorylation does not impact CaMKII activity in vitro. Although the role of these sites on molecular targeting of CaMKII has not been investigated, it is possible that, along with Thr253, they may belong to a class of phosphorylation site that has its major functional role in regulating CaMKII targeting rather than directly altering kinase activity.

### 3.2. Targeting

The importance, and mechanisms, of targeting in the regulation of CaMKII has recently begun to be appreciated.

#### 3.2.1. Splice Isoforms

All four CaMKII genes undergo alternative splicing [51], which can produce changes in specific activity, maximal autonomy and differential responses to calcium oscillations in vitro [66]. Perhaps one of the most unusual examples of the splicing of CaMKII - producing alterations in targeting - is the αCaMKII-anchoring protein (αKAP). This truncated, enzymatically inactive splice variant is only made up of the association domain and a lipid tail. αKAP is primarily expressed in muscle (skeletal and cardiac), and at low levels in the testis, lung and kidney [67]. The lipid tail allows anchoring and targeting to the sarcoplasmic reticulum (SR) membrane in skeletal muscle. Importantly, αKAP forms heteromultimers with full-length CaMKII, which facilitates the translocation of active full-length kinase subunits to the SR membrane, which would not normally occur [68]. Additionally, a small number of splice variants, such as the αB subunit, contain a nuclear localisation sequence (NLS) which allows the targeting of CaMKII to the nucleus [69]. Other isoforms, such as CaMKIIβ, contain specific binding sites for individual proteins (specifically F-actin) [70,71]. Additionally, the small C-terminal domain phosphatase 3 (SCP3), a PP2C-type phosphatase, binds to the association domain of the G-2 variant of CaMKIIγ, thereby selectively dephosphorylating it [72].

The importance of targeting and isoforms to the regulation of CaMKII is particularly highlighted in CaMKII-mediated cell proliferation. The inhibition of CaMKII can inhibit proliferation in a plethora of cell types [73,74,75]. Of note, in vascular smooth muscle cells, serum withdrawal induces expression of the CaMKII∂2 splice variant, resulting in an increase in proliferation. Conversely, this pro-proliferative effect can be inhibited by downregulating the CaMKII∂2 isoform [73]. Further, in rabbit reticulocyte lysates, expression of an autonomously active truncated form of CaMKII, where everything from the middle of the calmodulin binding domain to the C-terminal end was removed (thereby eliminating most, if not all, of the proposed targeting sequences), inhibited proliferation [76].

#### 3.2.2. Translocation

The oldest evidence in support of CaMKII targeting regulating function comes from the evidence that CaMKII is concentrated in particular subcellular locations, such as the PSD, and that CaMKII can translocate from the cytoplasm (reviewed in [77,78]). This occurs in response to a variety of stimuli including hypoxia, cellular excitation, and post-mortem delay, as well as during the normal maturation phase of brain development [79,80,81,82]. CaMKII autophosphorylation can also control translocation and CaMKII binding to the PSD (Table 1). For example, phosphorylation of Thr305/306 stimulates translocation from the PSD to the cytoplasm [83]. By contrast, phosphorylation of either Thr286 or Thr253 enhances binding to the PSD, and phosphorylation at both sites results in an additive effect [84], suggesting that these translocations are mediated by different binding proteins. Once located in the PSD, CaMKII can phosphorylate a variety of substrates, including NMDA and AMPA receptors [85,86,87,88].

CaMKII can also translocate to the cytoskeleton [89,90]. Binding to components of the cytoskeleton, such as MAP-2, is enhanced following Thr286 phosphorylation when compared to non-phosphorylated or Thr253 phosphorylated CaMKII [91,92,93]. However, the effects of phosphorylation of CaMKII on translocation to the cytoskeleton have not been characterised.

#### 3.2.3. Binding Proteins

Regarding CaMKII function, CaMKII must be co-localised with the correct binding partner to produce the required cellular function. A variety of binding proteins have been identified that bind to CaMKII in a manner that varies with the phosphorylation state of CaMKII (Table 2). Since there is a range of CaMKII binding proteins in cells that bind either non-phosphorylated or phosphorylated CaMKII, it is likely that most, if not all, CaMKII in cells is bound to a protein rather than free in the cytoplasm.

Targeting can achieve localised, or cell-specific, functions in one of two ways (shown schematically in Figure 2). When a rise in intracellular calcium activates CaMKII and results in autophosphorylation (①), the binding proteins that were associated with the inactive CaMKII dissociate (②) and the phosphorylated CaMKII binds to another (the brown oblong) binding protein (③). This binding protein can direct the kinase activity of CaMKII (④) to substrates located nearby (e.g., the blue oblongs)—or alternatively, the binding protein itself—resulting in a selective or quicker phosphorylation of specific proteins after a future rise in intracellular calcium (⑤). Through allosteric mechanisms, interactions with some binding partners can activate CaMKII without the need for a rise in intracellular calcium, which can enable CaMKII to phosphorylate substrates in vivo that are only poorly phosphorylated in vitro [85,94,95]. Alternatively, binding proteins can target the adaptor activity of CaMKII (⑥) to recruit other proteins (⑦) to form a signalling complex. However, this may not necessarily result in the phosphorylation of any of the proteins within the signalling complex. The targeting of kinase and adaptor activities can occur either together or independently. For clarity, the schematic diagram in Figure 2 depicts these two activities acting independently.

**Table 2 cells-12-00401-t002:** Proteins that alter their binding to CaMKII in response to changes in CaMKII phosphorylation.

Binding Partner		Effect of CaMKII Phosphorylation/Phospho-Mimic Mutation on Binding	Ref
Protein	Cellular Function		
Brain and acute leukaemia (BAALC) 1-6-8	Haematopoietic cell proliferation, survival, and differentiation	Non-phosphorylated and 286 phospho-mimic mutants bind, 253 phospho-mimic mutation significantly increases binding	[93,96]
Calcium channel α-subunit isoforms (L-type)	Calcium influx	Non-phosphorylated CaMKII binds α1, α2a, α3 and α4 subunits; pThr286 binds only α1 and α2a subunits	[97]
Camguk/CASK	Synaptic targeting and synaptic plasticity	Non-phosphorylated CaMKII binds, pThr305/306 decreases binding	[95]
Densin-180 (LRRC7)	Dendritic scaffolding protein	pThr286 enhances binding compared to non-phosphorylated CaMKII	[98,99]
Desmin	Muscle intermediate filament	Non-phosphorylated CaMKII binds, 286 and 253 phospho-mimic mutations increase binding	[91,92,93]
Diacylglycerol lipase α (DGLα)	Key enzyme in biosynthesis of the endocannabinoid 2-arachidonoylglycerol	Non-phosphorylated CaMKII does not bind, pT286 directly interacts	[100]
Metabotropic glutamate receptor 5 (mGluR5)	Metabotropic glutamate receptor involved in various brain functions, including motor behaviour, memory and cognition	pT286 decreases binding compared to non-phosphorylated CaMKII	[101]
Microtubule-associated protein 2 (MAP-2)	Microtubule assembly	Non-phosphorylated CaMKII binds, 286 phospho-mimic mutation increases binding	[91,92,93]
Myelin Basic Protein (MBP)	Major component of the myelin sheath	Non-phosphorylated CaMKII binds, 286 phospho-mimic mutation slightly decreases binding, 253 phospho-mimic mutation abrogates binding	[93]
GRIN2A/2B (GluN2A/2B)	Subunit of voltage-sensitive ionotropic glutamate receptor involved in synaptic plasticity	pThr286 enhances binding compared to non-phosphorylated CaMKII	[85,86,87,88]
Projectin	Integral protein of insect flight muscle	pThr286 decreases binding compared to non-phosphorylated CaMKII	[102]
Syntaxin 1A	Component of exocytotic molecular machinery	Non-phosphorylated CaMKII did not bind, pT286 directly interacts	[103]
Tau	Microtubule assembly	Non-phosphorylated and 253 phospho-mimic mutation bind, 286 phospho-mimic mutation increases binding	[92,93,104]
Tyrosine hydroxylase isoform 2 (human)	Catecholamine biosynthesis	Non-phosphorylated and 286 phospho-mimic mutants bind, 253 phospho-mimic mutation increases binding	[93]
Tyrosine hydroxylase isoform 2 (human), phosphorylated at Ser19 and Ser40	Catecholamine biosynthesis	Binding is enhanced for non-phosphorylated, 286 and 253 phospho-mimic mutation when compared to non-phosphorylated tyrosine hydroxylase	[93]

In addition to the proteins in Table 2, a wider range of proteins (reviewed in [105]), including enzymes, ion channels and cytoskeletal proteins, have also been shown to bind CaMKII, but the effects of phosphorylation on the binding of those proteins have not yet been reported.

#### 3.2.4. Phosphatases

Protein phosphatases (PP) can regulate CaMKII, via actions on both CaMKII and its binding partners. PP1, PP2A and PP2C, but not PP2B, can dephosphorylate Thr286 in vitro [106,107,108]. Additionally, PP2A, but not PP1, can dephosphorylate Thr253 in cancer cells in vitro [23].

While PSD-associated CaMKII is primarily dephosphorylated by PP1 [80,106,108,109], Thr286 phosphorylation appears to be protected from dephosphorylation in this case [110]. By contrast, cytoplasmic CaMKII appears to be dephosphorylated by PP2A [80]. Therefore, CaMKII located in different subcellular locations may be exposed to distinct phosphatases or differing levels of phosphatase activity, thus further highlighting the importance of the cellular microenvironment in the regulation of CaMKII in vivo.

## 4. Molecular and Cellular Micro-Environment

Changes caused by the molecular/cellular environment, largely due to the effects of CaMKII binding proteins, can bias the CaMKII autophosphorylation response to activation by calcium/calmodulin. The pattern of expression of CaMKII binding proteins varies with cell type and between nerve cells in different brain regions [93]. Such cell-specific variation in the expression of CaMKII binding proteins is involved in creating populations of CaMKII with different molecular environments that produce different functional responses to CaMKII activation. Depending on the binding protein, CaMKII can (i) become selectively responsive to stimuli that activate particular calcium channels by localising CaMKII to those channels (e.g., [85]); (ii) the binding protein can induce conformational changes in CaMKII to alter its activity or favour autophosphorylation at one or more sites (e.g., [95,111]); or (iii) by locating its kinase activity near particular substrates CaMKII can selectively regulate molecular pathways (e.g., [112]) thereby producing different functional responses to activation (Figure 3).

Figure 3 shows a schematic diagram of how such differences in CaMKII-mediated functional responses can be produced by differences in the molecular environment. Three populations of CaMKII are shown, which could be in different cell types or in different subcellular locations within the same cell. The three populations of CaMKII are bound to different proteins that each contain a binding site for dephosphorylated CaMKII (Figure 3A). A rise in intracellular calcium following stimulation of the cell activates CaMKII but results in different patterns of autophosphorylation in the three populations of CaMKII due to the effects described above. The different patterns of autophosphorylation cause different allosteric changes in CaMKII exposing new phosphorylation-state-specific protein binding sites and causing CaMKII to dissociate from the proteins that bound dephosphorylated CaMKII (Figure 3B). The three different phosphorylated CaMKII populations combine with their own specific binding proteins which link to different molecular pathways, and/or translocate the CaMKII to a different intracellular site, thereby resulting in different functional outcomes from the original CaMKII activation (Figure 3C). These pools of CaMKII may be activated in their new location by calcium/calmodulin following a subsequent rise in intracellular calcium, or may become autonomously active due to Thr286 phosphorylation or allosteric activation of the targeting protein and/or other proteins in the cellular microenvironment [16,17].

Additionally, the molecular environment can produce differential CaMKII functional responses by the stimulus-induced movement of CaMKII between different binding partners in protein complexes, such as those found in the PSD or the cytoskeleton (Figure 4). Subsequent stimulus-induced rises in intracellular calcium can induce the translocation of CaMKII, which can activate CaMKII, or other signalling pathways, in different ways to induce different CaMKII-mediated cellular responses.

Panel A of Figure 4 shows non-phosphorylated CaMKII bound to a protein in a molecular environment containing several CaMKII binding proteins in a membrane-associated protein complex. A stimulus-induced rise in intracellular calcium activates the non-phosphorylated CaMKII with calcium/calmodulin allowing CaMKII to phosphorylate itself and nearby substrates in the protein complex (shown as response ①). The phosphorylated-CaMKII undergoes an allosteric change that causes it to dissociate from its previous binding protein, exposes a new binding site on its surface, and allows it to translocate and bind to a nearby protein that has the binding site specific to the phosphorylated CaMKII. When a second stimulus-induced rise in intracellular calcium activates the translocated CaMKII, it phosphorylates two different sites on neighbouring proteins (shown as response ②). This means that two stimuli of the same type can have different CaMKII-mediated functional outcomes.

Panel B shows how changes caused by a different signalling pathway can also alter CaMKII-mediated responses. At the top, non-phosphorylated CaMKII is shown bound to the same protein, in the same molecular environment, as in Panel A. A stimulus that does not involve CaMKII is shown resulting in the phosphorylation of a CaMKII binding protein in the complex that has a binding site specific for phosphorylated CaMKII and at a site that inhibits the protein’s ability to bind phosphorylated CaMKII. The initial stimulus-induced rise in intracellular calcium shown in Panel A allows the non-phosphorylated CaMKII to be activated by calcium/calmodulin and to produce the functional response ① (Figure 4A). However, when the phosphorylated-CaMKII dissociates from the protein to which non-phosphorylated CaMKII was bound, its ability to translocate to the same phospho-CaMKII binding protein as in A, has been blocked by the phosphorylation of this protein following stimulation of the different signalling pathway at the top of Panel B. Consequently, the phosphorylated CaMKII translocates to another binding protein in a nearby complex that also has a binding site specific for the phosphorylated-CaMKII. This translocated phosphorylated-CaMKII, which is in a different immediate molecular environment, responds to the second stimulus-induced rise in intracellular calcium by phosphorylating a neighbouring protein in its new environment (shown as response ③). This illustrates how the same stimulus can produce different CaMKII-mediated functional outcomes depending on the prior stimulus-induced translocation of CaMKII and cross talk between different signalling pathways.

Comparisons of the CaMKII binding profiles of extracts from different cell lines, and tissues from different brain regions, revealed that different cell types express different CaMKII binding proteins and that treatment of tissue extracts with exogenous protein phosphatase greatly reduced CaMKIIα binding to many, though not all, of the proteins [93]. It is well established that CaMKII located in different compartments behaves differently (reviewed in [78]), due largely to the ability of binding partners to alter the phosphorylation state and function of CaMKII. Two binding proteins—the NMDA receptor and CASK—which are both concentrated at the PSD, illustrate these reciprocal interactions.

Once the NMDA receptor has been activated by synaptic activity, opening its calcium channel and raising intracellular calcium concentrations in the post-synaptic spine and local dendrite, CaMKII in the vicinity can be activated by calcium/calmodulin. The resulting autophosphorylation of Thr286 allows CaMKII to bind to the GluN2B subunit of the NMDA receptor [85,94]. As long as the CaMKII remains associated with GluN2B, the conformational change induced in CaMKII maintains its autonomous activity even if CaMKII is dephosphorylated at Thr286 and if calmodulin dissociates following a drop in intracellular calcium levels. Furthermore, the phosphorylation of Thr305/306 that would occur in autonomously active CaMKII in vitro, is inhibited. This permits the CaMKII bound to GluN2B to bind calcium/calmodulin again following a subsequent increase in intracellular calcium concentrations.

When the scaffold protein CASK binds to non-phosphorylated CaMKII, the interaction also induces a conformational change in CaMKII that autonomously activates it, but in a way that favours autophosphorylation at Thr305/306 [95]. When intracellular calcium levels fall allowing calmodulin to dissociate from its binding site and reveal Thr305/306, the CASK-induced autonomous activity allows CaMKII to autophosphorylate at Thr305/306. This phosphorylation disrupts the binding to CASK, releasing a pool of CaMKII that cannot bind calcium/calmodulin and, therefore, is insensitive to changes in intracellular calcium levels until phosphatase activity can restore its ability to bind calcium/calmodulin. This may provide a mechanism for differentiating between inactive and active synapses by downregulating the pool of CaMKII capable of being activated by increases in intracellular calcium [95].

Another example of how stimulation-induced changes in interacting proteins can alter the functional outcomes of CaMKII activation is provided by the mechanisms of induction of LTP (long-term potentiation) and LTD (long-term depression)—two opposing forms of synaptic plasticity involved in learning, memory and cognition. DAPK1 (Death Associated Protein Kinase 1) is a calcium/calmodulin-dependent protein kinase that binds to the same region of the GluN2B subunit of the NMDA receptor as CaMKII, and, like CaMKII, is basally concentrated at dendritic spines. Patterns of synaptic stimulation that produce LTP induce high levels of Ca^2+^ influx into the dendritic spine activate CaMKII and autophosphorylation at Thr286. It also displaces DAPK1 that is bound to GluN2B and allows p-Thr286-CaMKII to bind to GluN2B, causing CaMKII to accumulate at the potentiated synapses. By contrast, the patterns of synaptic stimulation that produce LTD induce relatively lower levels of Ca^2+^ influx into the dendritic spine. This leads to calcineurin-mediated activation of DAPK1 and enhanced binding of DAPK1 to GluNR2B which, in turn, prevents the binding of CaMKII to GluN2B and CaMKII accumulation at depressed synapses [113].

Table 2 shows the variety of ways in which the interactions between CaMKII and its binding proteins can be modified by changes in the phosphorylation of CaMKII or the binding protein. The effects of CaMKII autophosphorylation at its different sites independently affect interactions with the binding proteins. For example, phosphorylation at Thr286 increases CaMKIIα binding to several microtubule-associated proteins while phosphorylation at Thr253 has little or no effect. By contrast, for both BAALC 1-6-8, a membrane targeting protein [96] and the enzyme TH2 (human tyrosine hydroxylase isoform 2), Thr253 phosphorylation increases CaMKIIα binding [93]. However, phosphorylation of TH2 at Ser19 and Ser40 greatly enhanced the affinity of TH2 binding by all forms of CaMKIIα and magnified the relative enhancement of binding by Thr253 phosphorylation. The Thr253Asp phospho-mimic mutation had no effect on the kinetics of TH2 phosphorylation by CaMKII [84] indicating that, for its interaction with TH2, phosphorylation at Thr253 specifically modifies the CaMKII’s adaptor activity (Figure 2).

Because CaMKII is involved in so many different cellular functions, including several pathological conditions, the ability to inhibit selectively some of these CaMKII-mediated functions offers potential therapeutic opportunities. However, precisely because CaMKII is expressed in so many different cells, direct inhibition of CaMKII kinase activity has been dismissed as unlikely to be therapeutically useful because of the high likelihood of unwanted side effects due to the inhibition of CaMKII in tissues, or functional responses, which are not involved in the pathological process that is the target of the desired therapeutic drug. The ability to indirectly inhibit CaMKII-mediated functional responses by specifically inhibiting the molecular targeting mechanism that controls the cell-specific CaMKII functional response offers new possibilities for the development of pharmaceutical agents with potential clinical application (Figure 5). Two populations of CaMKII, which could be in different cell types or different locations in the same cell, are shown to be activated by a rise in intracellular calcium and undergo different patterns of autophosphorylation, which would normally lead to binding to different CaMKII binding proteins. In the presence of a low molecular weight inhibitor with a high affinity for the binding site on the targeting protein for Thr253-CaMKII, the interaction between the binding protein and Thr253-CaMKII cannot occur, and the functional response is blocked. Since the inhibitor is specific for the targeting of Thr253-CaMKII, the functional response to the activation of the other population of CaMKII is unaffected. The same principle could be applied using an inhibitor that is specific for the binding site on the targeting protein for CaMKII phosphorylated at other sites, or on targeting proteins specifically expressed in particular cell types, to block CaMKII-mediated functions in that cell type.

Two examples of pathological conditions in which such specific inhibition of CaMKII targeting might have therapeutic application are neuronal cell death following a stroke and cell proliferation in cancer.

Regions of the brain that are more sensitive to ischaemic damage show an increased phosphorylation of CaMKII at Thr253 following ischaemia in vivo, or excitotoxicity in vitro, but no difference in phosphorylation at Thr286 or Thr305/306 [17]. CaMKII phosphorylation at Thr253 is essential for ischaemia-induced, CaMKII-mediated cell death [16]. The brain regions that are more susceptible to ischaemic damage also express patterns of CaMKII binding proteins that are different from those of the regions that are relatively resistant to ischaemia [93]. The binding of one or more of these proteins to pThr253-CaMKII induces a conformational change that activates CaMKII [16,17] and targets it to signalling complexes that activate cell death pathways. This provides an opportunity for therapeutic intervention with an inhibitor capable of mimicking the pThr253-induced binding site on CaMKII. Such an inhibitor could selectively prevent pThr253-CaMKII from binding to the specific protein that targets it to cell death pathways, thus reducing ischaemia-induced cell death without disrupting other CaMKII-mediated functional responses.

We have shown that phosphorylation of CaMKII at Thr286, but not Thr253, controls cancer cell migration and invasion [22], while phosphorylation at Thr253, but not Thr286, is essential for cancer cell proliferation [23]. Due to increased expression of CaMKII in a range of cancer types, and its importance in these cancer-related functions, there has been substantial interest in CaMKII as a potential anti-cancer drug target [21]. Whilst in vivo studies examining inhibitors of CaMKII activity, such as KN-93 and KN-62, have shown promising pre-clinical activity [21], due to the importance of CaMKII in a variety of brain and heart-related functions, the use of these inhibitors of CaMKII activity is likely to produce a plethora of side effects clinically. However, if the binding partners of pThr253-CaMKII that control cancer cell proliferation, and of pThr286-CaMKII responsible for cancer cell metastasis, can be identified, and inhibitors that selectively block these interactions developed, cancer cell proliferation or metastasis could be selectively inhibited, while leaving the normal brain and heart cell function unimpacted.

## 5. Future Directions

CaMKII located in different microenvironments in vivo can respond to stimuli differently—becoming phosphorylated at different sites resulting in differential interactions with binding proteins, translocation to different cellular locations and producing different functional outcomes. Future investigations should focus on characterising the molecular interactions responsible for the targeting of CaMKII in different cells in vivo. The structural features of the binding sites through which the targeting occurs could then be used to design inhibitors capable of selectively disrupting the interactions between CaMKII and specific binding partners in particular cells thereby interfering with the signalling pathways involved. This approach may then allow the selective control of the functions mediated by this multitasking protein in particular cells, with potential therapeutic applications in many clinical conditions such as ischaemic cell death following stroke [17].

## Figures and Tables

**Figure 2 cells-12-00401-f002:**
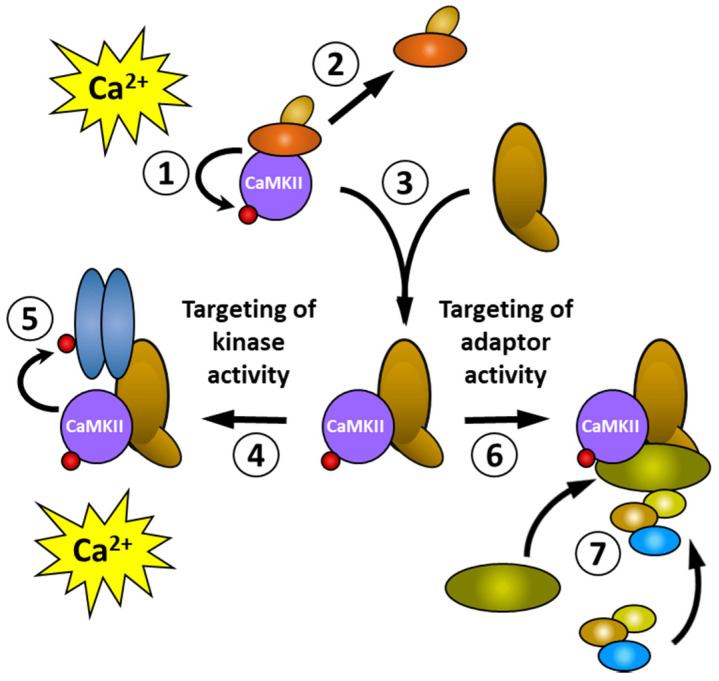
Targeting can produce localised functional outcomes by altering the kinase or adaptor activity of CaMKII. Coloured oblong shapes represent generic proteins with which CaMKII (purple disc) binds directly or indirectly. The small red disc represents the phosphorylation of a protein. See text for explanation of the sequence of events shown in the Figure.

**Figure 3 cells-12-00401-f003:**
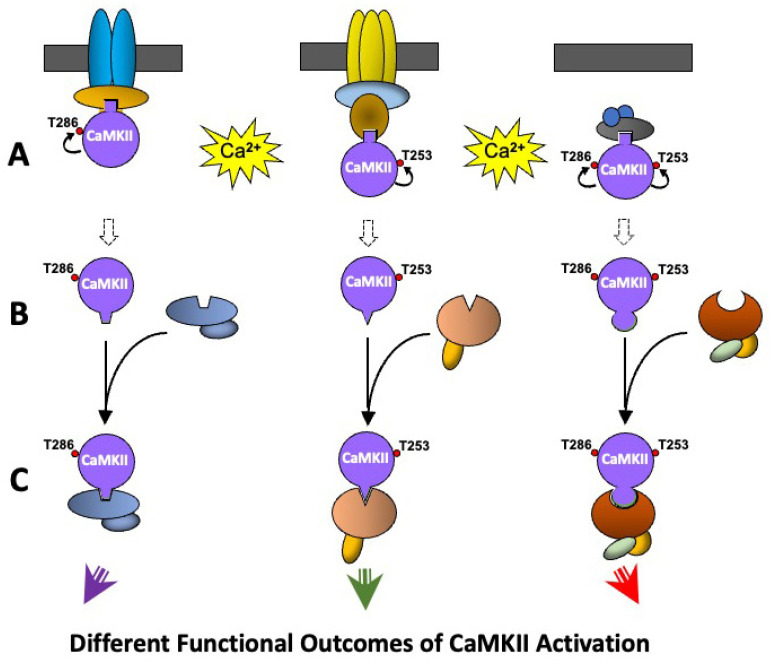
A schematic representation of how CaMKII in different molecular environments can respond to stimuli differently. The binding partner with which CaMKII is associated can influence the functional outcomes that occur following a rise in intracellular calcium. Coloured oblong shapes represent generic proteins with which CaMKII (purple disc) binds directly or indirectly. Dark Grey bar represents a membrane. Geometric protrusions from CaMKII, and indents in the binding proteins represent binding sites for CaMKII. The small red disc represents the phosphorylation of a protein. See text for explanation of the sequence of events shown in the Figure.

**Figure 4 cells-12-00401-f004:**
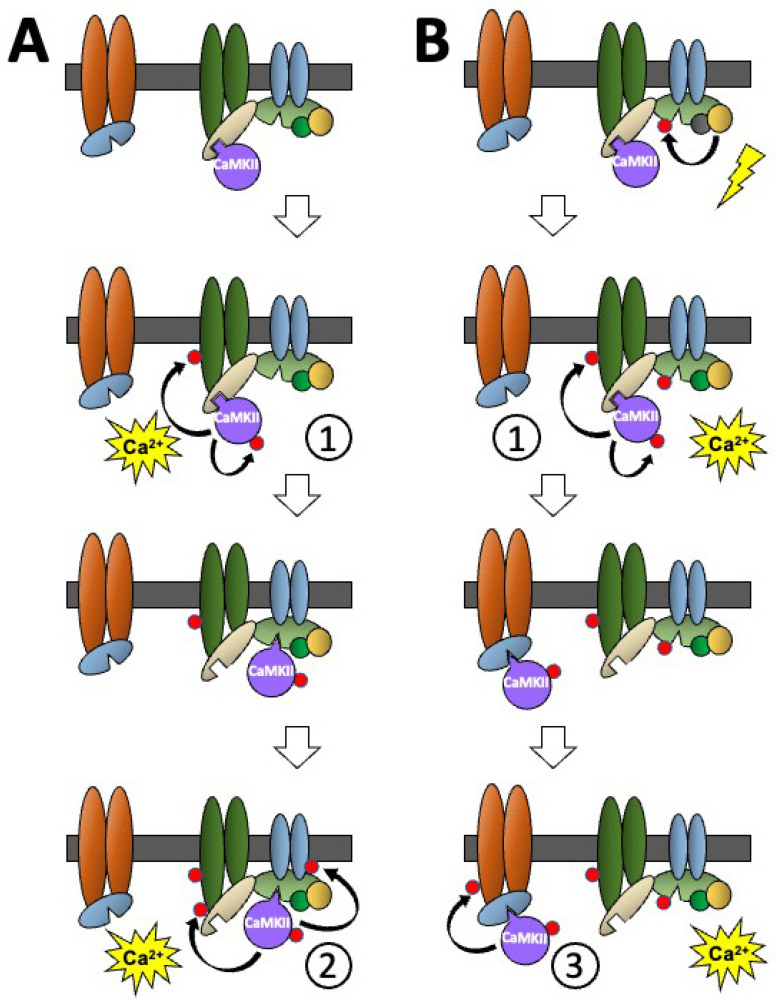
A schematic diagram showing how stimulus-induced translocation of CaMKII between different binding proteins in a molecular environment can produce different functional responses to the activation of CaMKII by two sequential stimuli of the same type and how the influence of another signalling pathway can further modify the CaMKII response. The effects of two sequential stimulus-induced rises in calcium on non-phosphorylated CaMKII, when the surrounding binding partners are (**A**) also non-phosphorylated, or (**B**) phosphorylated. Coloured oblong shapes represent generic proteins with which CaMKII (purple disc) binds directly or indirectly. Dark grey bar represents a membrane. Geometric protrusions from CaMKII, and indents in the binding proteins represent binding sites for CaMKII. The small red disc represents the phosphorylation of a protein. See text for explanation of the sequence of events shown in the Figure.

**Figure 5 cells-12-00401-f005:**
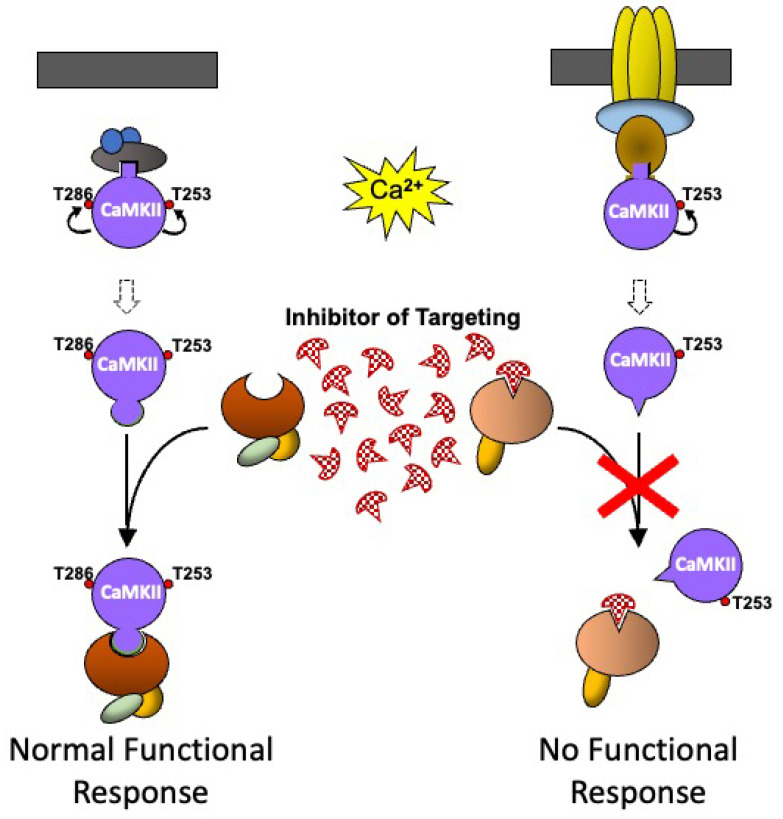
A schematic representation of how pharmacological agents that inhibit CaMKII targeting could be used therapeutically. As CaMKII that is phosphorylated at different sites binds to different binding proteins to produce different functional responses, an inhibitor that specifically blocks the interaction of CaMKII with a particular binding partner will inhibit a specific functional response, while leaving all the other pools of CaMKII bound to other binding proteins, untouched, and able to produce their normal functional response. Coloured oblong shapes represent generic proteins with which CaMKII (purple disc) binds directly or indirectly. Dark grey bar represents a membrane. Geometric protrusions from CaMKII, and indents in the binding proteins represent binding sites for CaMKII. The small red disc represents the phosphorylation of a protein. See text for explanation of the sequence of events shown in the Figure.

**Table 1 cells-12-00401-t001:** Effects on CaMKII activity and targeting following autophosphorylation at threonine (Thr) 253, 286 and 305/306.

Site of Autophosphorylation	Effect on CaMKII In Vitro			Effect on Targeting to:	
	Ca^2+^/CaM Binding	Ca^2+^/CaM Dependent Activity	Ca^2+^/CaM Independent Activity	Translocation to PSD ^#^ In Vivo ^#^	Binding to Specific Proteins
Thr286	↑	Prolonged	↑	↑	Multiple
Thr305/306	↓	↓	-	↓	N/E
Thr253	-	-	-	↑	Multiple

^#^ The post-synaptic density (PSD) is an assemblage of regulatory and cytoskeletal proteins on the cytoplasmic surface of the post-synaptic plasma membrane that regulates synaptic function. - = no effect; N/E = not examined.

## Data Availability

Not applicable.

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
