# Peer review of "Calcium/Calmodulin-Stimulated Protein Kinase II (CaMKII): Different Functional Outcomes from Activation, Depending on the Cellular Microenvironment"

_cells, 2023, doi:10.3390/cells12030401_

Round 1
Reviewer 1 Report
The manuscript by Rostas and Skelding presents a review of the structure and regulation of CaMKII that focuses on how different molecular stimuli can produce 16 different functional responses. Overall, the review provides important information but the presentation and organization needs some improvement. See my suggestions below that might help improve the manuscript.
- The section on CaMKII structure (section 2) is vague and needs to be more detailed with more references. Figure 1 could be improved by showing a more detailed atomic-resolution view of the oligomeric structure in panel C (eg see Figure 3A in Bayer and Schulman 2019). The structure of the holoenzyme is poorly described in section 2 and the description of this structure should be expanded. For example, the holoenzyme is known to exist in two forms: activation-competent extended structure and inactive compact structure as described by Myers et al (2017) Nat. Commun., 8, p. 15742. It would be good to include an illustration or schematic showing the two structural forms (extended and closed) that could be added to Figure 1.
- Figure 1C depicts the holoenzyme as a 12-mer. While the majority of the CaMKII holoenzyme forms a 12-mer in the cryo-EM images, a detectable fraction of the holoenzyme also exists as a 14-mer. There should be some discussion about the possibility that the 14-mer may represent an intermediate state that may mediate subunit exchange as discussed by Bhattacharyya et al., 2016, Rosenberg et al., 2006 and Stratton et al., 2014.
- The last paragraph of section 2 mentions that “Controversy has surrounded the precise structure of CaMKII”. It is not clear what is meant here by the term “controversy”. This paragraph is poorly written and needs to have more clarity. For example, it might help to add figures to illustrate the cryo-EM images of the association domain forming an annulus, the hub-and-spoke model, and structure from SAXS that is comprised of a symmetric dimer of two autoinhibited catalytic domains etc. It is not clear how there is a “controversy” here and there needs to be a clearer description of the holoenzyme in Fig. 1C and how it is regulated by calmodulin
- The interaction of CaMKII with calmodulin in Fig. 1C is too crude and does not provide a sufficiently accurate depiction of the regulatory mechanism. I suggest adding a figure that shows the amino acid sequence of the CaMKII regulatory domain that interacts with calmodulin, including auto-inhibition and CaM-binding regions and show sites that are posttranslationally modified by phosphorylation, nitrosylation and glycosylation. Also, it would be good to show a figure of the structural transition of the CaMKII regulatory domain in response to Ca2+/CaM binding as described by Rellos et al 2010 (eg see Fig. 2C in Bayer and Schulman 2019).
- Figures 2-5 legends are poorly constructed. Each legend fails to define the red circle, orange oval, dark blue pair of ovals, light-blue oval, yellow oval etc. Each legend needs a glossary of the different colored shapes to define which proteins or molecules are represented in each case.
- I suggest adding NMDA receptors and related references to Table 2. CaMKII binds to the C-terminal region of GluN2B subunit during LTP.
- The paragraphs on CaMKII interaction with NMDA receptors (pages 13-14) could be improved by adding a figure to illustrate the mechanism of CaMKII binding to GluN2B and its relation to LTP vs LTD (see figure in Goodell et al 2017). For example, GluN2B binds to CaMKII at high Ca2+ level to produce LTD, whereas GluN2B binds to DAPK1 at low Ca2+ levels to produce LTD.
Author Response
Authors’ Response to Reviewer 1
The manuscript by Rostas and Skelding presents a review of the structure and regulation of CaMKII that focuses on how different molecular stimuli can produce 16 different functional responses. Overall, the review provides important information but the presentation and organization needs some improvement. See my suggestions below that might help improve the manuscript.
- The section on CaMKII structure (section 2) is vague and needs to be more detailed with more references. Figure 1 could be improved by showing a more detailed atomic-resolution view of the oligomeric structure in panel C (eg see Figure 3A in Bayer and Schulman 2019). The structure of the holoenzyme is poorly described in section 2 and the description of this structure should be expanded. For example, the holoenzyme is known to exist in two forms: activation-competent extended structure and inactive compact structure as described by Myers et al (2017) Nat. Commun., 8, p. 15742. It would be good to include an illustration or schematic showing the two structural forms (extended and closed) that could be added to Figure 1.
- Figure 1C depicts the holoenzyme as a 12-mer. While the majority of the CaMKII holoenzyme forms a 12-mer in the cryo-EM images, a detectable fraction of the holoenzyme also exists as a 14-mer. There should be some discussion about the possibility that the 14-mer may represent an intermediate state that may mediate subunit exchange as discussed by Bhattacharyya et al., 2016, Rosenberg et al., 2006 and Stratton et al., 2014.
We have revised and expanded Section 2 to include the multiple forms in which CaMKII has been shown to exist, the equilibrium between the 12-mer and 14-mer structures, its role in subunit exchange and the proposed functions in which these equilibria are involved. The references to Myers et al (2017), Bhattacharyya et al (2016), Rosenberg et al (2006) and Stratton et al (2014) have been added.
We have not added another diagram to include an atomic-resolution view of the oligomeric structure of CaMKII in Figure 1, as a very similar figure has already published by other authors. Instead we have added a reference to the published paper containing the diagram.
- The last paragraph of section 2 mentions that “Controversy has surrounded the precise structure of CaMKII”. It is not clear what is meant here by the term “controversy”. This paragraph is poorly written and needs to have more clarity. For example, it might help to add figures to illustrate the cryo-EM images of the association domain forming an annulus, the hub-and-spoke model, and structure from SAXS that is comprised of a symmetric dimer of two autoinhibited catalytic domains etc. It is not clear how there is a “controversy” here and there needs to be a clearer description of the holoenzyme in Fig. 1C and how it is regulated by calmodulin.
The term “controversy” has been removed and the matters described in the original paragraph have been revised and incorporated into the new Section 2.
- The interaction of CaMKII with calmodulin in Fig. 1C is too crude and does not provide a sufficiently accurate depiction of the regulatory mechanism. I suggest adding a figure that shows the amino acid sequence of the CaMKII regulatory domain that interacts with calmodulin, including auto-inhibition and CaM-binding regions and show sites that are posttranslationally modified by phosphorylation, nitrosylation and glycosylation. Also, it would be good to show a figure of the structural transition of the CaMKII regulatory domain in response to Ca2+/CaM binding as described by Rellos et al 2010 (eg see Fig. 2C in Bayer and Schulman 2019).
The focus of this review is to examine the mechanisms by which different cellular microenvironments can regulate multiple functions of CaMKII and the role of binding proteins in these mechanisms. As a background to a discussion of the regulatory roles of binding proteins and cell microenvironments, an overview of the molecular structure of CaMKII is necessary but a detailed explanation of the molecular structure of CaMKII and its transitions under different circumstances is not necessary. For readers interested in the detailed structural changes in CaMKII, we have included references to appropriate papers. Also, the figures suggested by the reviewer are already published. Therefore, we have not included an extra figure.
- Figures 2-5 legends are poorly constructed. Each legend fails to define the red circle, orange oval, dark blue pair of ovals, light-blue oval, yellow oval etc. Each legend needs a glossary of the different colored shapes to define which proteins or molecules are represented in each case.
We have revised the legends to Figures 2-5 to make clear what each of the coloured shapes represent. The reviewer appears to have thought that the oblongs representing binding proteins were intended to depict specific binding proteins. In the revised figure legends we have clarified that the diagrams depict generic binding proteins acting in different ways.
- I suggest adding NMDA receptors and related references to Table 2. CaMKII binds to the C-terminal region of GluN2B subunit during LTP.
The NMDA receptor subunits were already included in Table 2 (GRIN2A/2B). To avoid any further confusion, we have added a note to the Table indicating that GRIN2A/2B is the gene for GluNR2A/2B.
- The paragraphs on CaMKII interaction with NMDA receptors (pages 13-14) could be improved by adding a figure to illustrate the mechanism of CaMKII binding to GluN2B and its relation to LTP vs LTD (see figure in Goodell et al 2017). For example, GluN2B binds to CaMKII at high Ca2+ level to produce LTD, whereas GluN2B binds to DAPK1 at low Ca2+ levels to produce LTD.
The focus of this review is how different CaMKII binding proteins in different cellular microenvironments can modify CaMKII activity and regulate different cellular functions. We believe that the two paragraphs describing the differences in CaMKII interaction with GluN2B and CASK illustrate how two binding proteins in the same cellular microenvironment can have very different effects on CaMKII. To provide a specific functional example, we have added a paragraph referring to the role of DAPK1-GluN2B interaction under different synaptic stimulation conditions and the effect of this on CaMKII binding to GluN2B.
Reviewer 2 Report
This is a good piece of research addressing an important topic and can be accepted for publication after addressing these points.
-
I feel there lacks a connection in the introduction section and it needs to be more connected. Adding a paragraph about protein kinases will be an added advantage and aid in the improvement. https://doi.org/10.1016/j.bbcan.2021.188568
-
Figure legends should be crisp.
-
Quality of Figure 2 needs to be enhanced.
-
Future directions need to be improved.
-
Aim of the study should be mentioned.
-
What was the rationale for this article as there are many other articles in a similar domain.
Author Response
Response to Reviewer 2
This is a good piece of research addressing an important topic and can be accepted for publication after addressing these points.
- I feel there lacks a connection in the introduction section and it needs to be more connected. Adding a paragraph about protein kinases will be an added advantage and aid in the improvement.
We have added an introductory paragraph about protein kinases at the beginning of the introduction.
- Figure legends should be crisp.
We have kept the figure legends as brief as possible (while still responding to Reviewer 1’s comments above) by referring the reader to the text for a detailed explanation of the events described in the figures.
- Quality of Figure 2 needs to be enhanced.
We revised Figure 2 and its explanation in the text to make it clearer.
- Future directions need to be improved.
The Reviewer does not indicate what sort of improvement was required to the Future Directions section. We have added a specific example of a clinical condition (ischaemic cell death following stroke) for which this approach promises considerable therapeutic potential.
- Aim of the study should be mentioned.
The aim of our review is indicated in the last paragraph of the introduction.
- What was the rationale for this article as there are many other articles in a similar domain.
We were invited to write a review on this topic for a special issue of Cells. While there are many other reviews of the role of CaMKII in different cellular processes, particularly in the nervous system, none of these specifically focus on the effects of the cellular microenvironment and CaMKII binding proteins in regulating different CaMKII-mediated functional outcomes.